# Learning the Morphology of Brain Signals Using Alpha-Stable Convolutional Sparse Coding

**Mainak Jas[1], Tom Dupré La Tour[1], Umut Şimşekli[1], Alexandre Gramfort[1,2]**
1: LTCI, Telecom ParisTech, Université Paris-Saclay, Paris, France
2: INRIA, Université Paris-Saclay, Saclay, France

## Abstract

Neural time-series data contain a wide variety of prototypical signal waveforms (atoms) that are of significant importance in clinical and cognitive research. One of the goals for analyzing such data is hence to extract such 'shift-invariant' atoms. Even though some success has been reported with existing algorithms, they are limited in applicability due to their heuristic nature. Moreover, they are often vulnerable to artifacts and impulsive noise, which are typically present in raw neural recordings. In this study, we address these issues and propose a novel probabilistic convolutional sparse coding (CSC) model for learning shift-invariant atoms from raw neural signals containing potentially severe artifacts. In the core of our model, which we call $\alpha$CSC, lies a family of heavy-tailed distributions called $\alpha$-stable distributions. We develop a novel, computationally efficient Monte Carlo expectation-maximization algorithm for inference. The maximization step boils down to a weighted CSC problem, for which we develop a computationally efficient optimization algorithm. Our results show that the proposed algorithm achieves state-of-the-art convergence speeds. Besides, $\alpha$CSC is significantly more robust to artifacts when compared to three competing algorithms: it can extract spike bursts, oscillations, and even reveal more subtle phenomena such as cross-frequency coupling when applied to noisy neural time series.

## 1  Introduction

Neural time series data, either non-invasive such as electroencephalograhy (EEG) or invasive such as electrocorticography (ECoG) and local field potentials (LFP), are fundamental to modern experimental neuroscience. Such recordings contain a wide variety of 'prototypical signals' that range from beta rhythms (12–30 Hz) in motor imagery tasks and alpha oscillations (8–12 Hz) involved in attention mechanisms, to spindles in sleep studies, and the classical P300 event related potential, a biomarker for surprise. These prototypical waveforms are considered critical in clinical and cognitive research [1], thereby motivating the development of computational tools for learning such signals from data.

Despite the underlying complexity in the morphology of neural signals, the majority of the computational tools in the community are based on representing the signals with rather simple, predefined bases, such as the Fourier or wavelet bases [2]. While such bases lead to computationally efficient algorithms, they often fall short at capturing the precise morphology of signal waveforms, as demonstrated by a number of recent studies [3, 4]. An example of such a failure is the disambiguation of the alpha rhythm from the mu rhythm [5], both of which have a component around 10 Hz but with different morphologies that cannot be captured by Fourier- or wavelet-based representations.

Recently, there have been several attempts for extracting more realistic and precise morphologies directly from unfiltered electrophysiology signals, via dictionary learning approaches [6–9]. These methods all aim to extract certain *shift-invariant* prototypical waveforms (called 'atoms' in this context) to better capture the temporal structure of the signals. As opposed to using generic bases

that have predefined shapes, such as the Fourier or the wavelet bases, these atoms provide a more meaningful representation of the data and are not restricted to narrow frequency bands.

In this line of research, Jost et al. [6] proposed the MoTIF algorithm, which uses an iterative strategy based on generalized eigenvalue decompositions, where the atoms are assumed to be orthogonal to each other and learnt one by one in a greedy way. More recently, the 'sliding window matching' (SWM) algorithm [9] was proposed for learning time-varying atoms by using a correlation-based approach that aims to identify the recurring patterns. Even though some success has been reported with these algorithms, they have several limitations: SWM uses a slow stochastic search inspired by simulated annealing and MoTIF poorly handles correlated atoms, simultaneously activated, or having varying amplitudes; some cases which often occur in practical applications.

A natural way to cast the problem of learning a dictionary of shift-invariant atoms into an optimization problem is a convolutional sparse coding (CSC) approach [10]. This approach has gained popularity in computer vision [11–15], biomedical imaging [16] and audio signal processing [10, 17], due to its ability to obtain compact representations of the signals and to incorporate the temporal structure of the signals via convolution. In the neuroscience context, Barthélemy et al. [18] used an extension of the K-SVD algorithm using convolutions on EEG data. In a similar spirit, Brockmeier and Príncipe [7] used the matching pursuit algorithm combined with a rather heuristic dictionary update, which is similar to the MoTIF algorithm. In a very recent study, Hitziger et al. [8] proposed the AWL algorithm, which presents a mathematically more principled CSC approach for modeling neural signals. Yet, as opposed to classical CSC approaches, the AWL algorithm imposes additional combinatorial constraints, which limit its scope to certain data that contain spike-like atoms. Also, since these constraints increase the complexity of the optimization problem, the authors had to resort to dataset-specific initializations and many heuristics in their inference procedure.

While the current state-of-the-art CSC methods have a strong potential for modeling neural signals, they might also be limited as they consider an $\ell_2$ reconstruction error, which corresponds to assuming an additive Gaussian noise distribution. While this assumption could be reasonable for several signal processing tasks, it turns out to be very restrictive for neural signals, which often contain heavy noise bursts and have low signal-to-noise ratio.

In this study, we aim to address the aforementioned concerns and propose a novel probabilistic CSC model called $\alpha$CSC, which is better-suited for neural signals. $\alpha$CSC is based on a family of *heavy-tailed* distributions called $\alpha$-stable distributions [19] whose rich structure covers a broad range of noise distributions. The heavy-tailed nature of the $\alpha$-stable distributions renders our model robust to impulsive observations. We develop a Monte Carlo expectation maximization (MCEM) algorithm for inference, with a weighted CSC model for the maximization step. We propose efficient optimization strategies that are specifically designed for neural time series. We illustrate the benefits of the proposed approach on both synthetic and real datasets.

## 2 Preliminaries

**Notation:** For a vector $v \in \mathbb{R}^n$ we denote the $\ell_p$ norm by $\|v\|_p = \left(\sum_i |v_i|^p\right)^{1/p}$. The convolution of two vectors $v_1 \in \mathbb{R}^N$ and $v_2 \in \mathbb{R}^M$ is denoted by $v_1 * v_2 \in \mathbb{R}^{N+M-1}$. We denote by $x$ the observed signals, $d$ the temporal atoms, and $z$ the sparse vector of *activations*. The symbols $\mathcal{U}, \mathcal{E}, \mathcal{N}, \mathcal{S}$ denote the univariate uniform, exponential, Gaussian, and $\alpha$-stable distributions, respectively.

**Convolutional sparse coding:** The CSC problem formulation adopted in this work follows the Shift Invariant Sparse Coding (SISC) model from [10]. It is defined as follows:

$$\min_{d,z} \sum_{n=1}^{N} \left( \frac{1}{2} \|x_n - \sum_{k=1}^{K} d^k * z_n^k\|_2^2 + \lambda \sum_{k=1}^{K} \|z_n^k\|_1 \right), \quad \text{s.t.} \quad \|d^k\|_2^2 \leq 1 \text{ and } z_n^k \geq 0, \forall n, k \ , \quad (1)$$

where $x_n \in \mathbb{R}^T$ denotes one of the $N$ observed segments of signals, also referred to as a *trials* in this paper. We denote by $T$ as the length of a trial, and $K$ the number of atoms. The aim in this model is to approximate the signals $x_n$ by the convolution of certain *atoms* and their respective *activations*, which are sparse. Here, $d^k \in \mathbb{R}^L$ denotes the $k$th atom of the *dictionary* $d \equiv \{d^k\}_k$, and $z_n^k \in \mathbb{R}_+^{T-L+1}$ denotes the activation of the $k$th atom in the $n$th trial. We denote by $z \equiv \{z_n^k\}_{n,k}$.

The objective function (1) has two terms, an $\ell_2$ data fitting term that corresponds to assuming an additive Gaussian noise model, and a regularization term that promotes sparsity with an $\ell_1$ norm. The

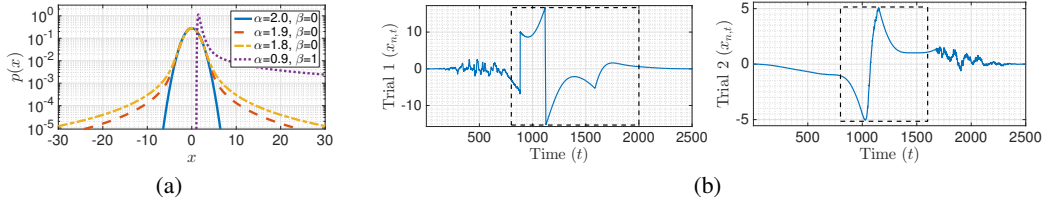

(a)                                                                (b)

Figure 1: (a) PDFs of $\alpha$-stable distributions. (b) Illustration of two trials from the striatal LFP data, which contain severe artifacts. The artifacts are illustrated with dashed rectangles.

regularization parameter is called $\lambda > 0$. Two constraints are also imposed. First, we ensure that $d^k$ lies within the unit sphere, which prevents the scale ambiguity between $d$ and $z$. Second, a positivity constraint on $z$ is imposed to be able to obtain physically meaningful activations and to avoid sign ambiguities between $d$ and $z$. This positivity constraint is not present in the original SISC model [10].

$\alpha$**-Stable distributions:** The $\alpha$-stable distributions have become increasingly popular in modeling signals that might incur large variations [20–24] and have a particular importance in statistics since they appear as the limiting distributions in the generalized central limit theorem [19]. They are characterized by four parameters: $\alpha$, $\beta$, $\sigma$, and $\mu$: (i) $\alpha \in (0, 2]$ is the *characteristic exponent* and determines the tail thickness of the distribution: the distribution will be heavier-tailed as $\alpha$ gets smaller. (ii) $\beta \in [-1, 1]$ is the *skewness* parameter. If $\beta = 0$, the distribution is symmetric. (iii) $\sigma \in (0, \infty)$ is the *scale* parameter and measures the spread of the random variable around its mode (similar to the standard deviation of a Gaussian distribution). Finally, (iv) $\mu \in (-\infty, \infty)$ is the location parameter (for $\alpha > 1$, it is simply the mean).

The probability density function of an $\alpha$-stable distribution cannot be written in closed-form except for certain special cases; however, the characteristic function can be written as follows:

$$x \sim \mathcal{S}(\alpha, \beta, \sigma, \mu) \iff \mathbb{E}[\exp(i\omega x)] = \exp(-|\sigma\omega|^\alpha [1 + i\,\text{sign}(\omega)\beta\psi_\alpha(\omega)] + i\mu\omega) \ ,$$

where $\psi_\alpha(\omega) = \log|\omega|$ for $\alpha = 1$, $\psi_\alpha(\omega) = \tan(\pi\alpha/2)$ for $\alpha \neq 1$, and $i = \sqrt{-1}$. As an important special case of the $\alpha$-stable distributions, we obtain the Gaussian distribution when $\alpha = 2$ and $\beta = 0$, *i.e.* $\mathcal{S}(2, 0, \sigma, \mu) = \mathcal{N}(\mu, 2\sigma^2)$. In Fig. 1(a), we illustrate the (approximately computed) probability density functions (PDF) of the $\alpha$-stable distribution for different values of $\alpha$ and $\beta$. The distribution becomes heavier-tailed as we decrease $\alpha$, whereas the tails vanish quickly when $\alpha = 2$.

The moments of the $\alpha$-stable distributions can only be defined up to the order $\alpha$, i.e. $\mathbb{E}[|x|^p] < \infty$ if and only if $p < \alpha$, which implies the distribution has infinite variance when $\alpha < 2$. Furthermore, despite the fact that the PDFs of $\alpha$-stable distributions do not admit an analytical form, it is straightforward to draw random samples from them [25].

## 3  Alpha-Stable Convolutional Sparse Coding

### 3.1  The Model

From a probabilistic perspective, the CSC problem can be also formulated as a maximum a-posteriori (MAP) estimation problem on the following probabilistic generative model:

$$z_{n,t}^k \sim \mathcal{E}(\lambda), \quad x_{n,t}|z, d \sim \mathcal{N}(\hat{x}_{n,t}, 1), \quad \text{where,} \quad \hat{x}_n \triangleq \sum_{k=1}^{K} d^k * z_n^k \ . \tag{2}$$

Here, $z_{n,t}^k$ denotes the $t$th element of $z_n^k$. We use the same notations for $x_{n,t}$ and $\hat{x}_{n,t}$. It is easy to verify that the MAP estimate for this probabilistic model, *i.e.* $\max_{d,z} \log p(d, z|x)$, is identical to the original optimization problem defined in (1)[1].

It has been long known that, due to their light-tailed nature, Gaussian models often fail at handling noisy high amplitude observations or outliers [26]. As a result, the 'vanilla' CSC model turns out to be highly sensitive to outliers and impulsive noise that frequently occur in electrophysiological

recordings, as illustrated in Fig. 1(b). Possible origins of such artifacts are movement, muscle contractions, ocular blinks or electrode contact losses.

In this study, we aim at developing a probabilistic CSC model that would be capable of modeling challenging electrophysiological signals. We propose an extension of the original CSC model defined in (2) by replacing the light-tailed Gaussian likelihood (corresponding to the $\ell_2$ reconstruction loss in (1)) with heavy-tailed $\alpha$-stable distributions. We define the proposed probabilistic model ($\alpha$CSC) as follows:

$$z_{n,t}^k \sim \mathcal{E}(\lambda), \quad x_{n,t}|z,d \sim \mathcal{S}(\alpha, 0, 1/\sqrt{2}, \hat{x}_{n,t}) \ , \tag{3}$$

where $\mathcal{S}$ denotes the $\alpha$-stable distribution. While still being able to capture the temporal structure of the observed signals via convolution, the proposed model has a richer structure and would allow large variations and outliers, thanks to the heavy-tailed $\alpha$-stable distributions. Note that the vanilla CSC defined in (2) appears as a special case of $\alpha$CSC, as the $\alpha$-stable distribution coincides with the Gaussian distribution when $\alpha = 2$.

## 3.2  Maximum A-Posteriori Inference

Given the observed signals $x$, we are interested in the MAP estimates, defined as follows:

$$(d^\star, z^\star) = \arg\max_{d,z} \sum_{n,t} \Big( \log p(x_{n,t}|d,z) + \sum_k \log p(z_{n,t}^k) \Big). \tag{4}$$

As opposed to the Gaussian case, unfortunately, this optimization problem is not amenable to classical optimization tools, since the PDF of the $\alpha$-stable distributions does not admit an analytical expression. As a remedy, we use the product property of the symmetric $\alpha$-stable densities [19, 27] and re-express the $\alpha$CSC model as conditionally Gaussian. It leads to:

$$z_{n,t}^k \sim \mathcal{E}(\lambda), \quad \phi_{n,t} \sim \mathcal{S}\Big(\frac{\alpha}{2}, 1, 2(\cos\frac{\pi\alpha}{4})^{2/\alpha}, 0\Big), \quad x_{n,t}|z,d,\phi \sim \mathcal{N}\Big(\hat{x}_{n,t}, \frac{1}{2}\phi_{n,t}\Big) \ , \tag{5}$$

where $\phi$ is called the *impulse* variable that is drawn from a *positive* $\alpha$-stable distribution (i.e. $\beta = 1$), whose PDF is illustrated in Fig. 1(a). It can be shown that both formulations of the $\alpha$CSC model are identical by marginalizing the joint distribution $p(x,d,z,\phi)$ over $\phi$ [19, Proposition 1.3.1].

The impulsive structure of the $\alpha$CSC model becomes more prominent in this formulation: the variances of the Gaussian observations are modulated by stable random variables with infinite variance, where the impulsiveness depends on the value of $\alpha$. It is also worth noting that when $\alpha = 2$, $\phi_{n,t}$ becomes deterministic and we can again verify that $\alpha$CSC coincides with the vanilla CSC.

The conditionally Gaussian structure of the augmented model has a crucial practical implication: if the impulse variable $\phi$ were to be known, then the MAP estimation problem over $d$ and $z$ in this model would turn into a 'weighted' CSC problem, which is a much easier task compared to the original problem. In order to be able to exploit this property, we propose an expectation-maximization (EM) algorithm, which iteratively maximizes a lower bound of the log-posterior $\log p(d,z|x)$, and algorithmically boils down to computing the following steps in an iterative manner:

$$\text{E-Step:} \qquad \mathcal{B}^{(i)}(d,z) = \mathbb{E}\left[\log p(x,\phi,z|d)\right]_{p(\phi|x,z^{(i)},d^{(i)})}, \tag{6}$$

$$\text{M-Step:} \qquad (d^{(i+1)}, z^{(i+1)}) = \arg\max_{d,z} \mathcal{B}^{(i)}(d,z). \tag{7}$$

where $\mathbb{E}[f(x)]_{q(x)}$ denotes the expectation of a function $f$ under the distribution $q$, $i$ denotes the iterations, and $\mathcal{B}^{(i)}$ is a lower bound to $\log p(d,z|x)$ and it is tight at the current iterates $z^{(i)}$, $d^{(i)}$.

**The E-Step:** In the first step of our algorithm, we need to compute the EM lower bound $\mathcal{B}$ that has the following form:

$$\mathcal{B}^{(i)}(d,z) =^+ -\sum_{n=1}^N \Big( \|\sqrt{w_n^{(i)}} \odot (x_n - \sum_{k=1}^K d^k * z_n^k)\|_2^2 + \lambda \sum_{k=1}^K \|z_n^k\|_1 \Big), \tag{8}$$

where $=^+$ denotes equality up to additive constants, $\odot$ denotes the Hadamard (element-wise) product, and the square-root operator is also defined element-wise. Here, $w_n^{(i)} \in \mathbb{R}_+^T$ are the *weights* that are defined as follows: $w_{n,t}^{(i)} \triangleq \mathbb{E}\left[1/\phi_{n,t}\right]_{p(\phi|x,z^{(i)},d^{(i)})}$. As the variables $\phi_{n,t}$ are expected to be large when $\hat{x}_{n,t}$ cannot explain the observation $x_{n,t}$ – typically due to a corruption or a high noise – the weights will accordingly suppress the importance of the particular point $x_{n,t}$. Therefore, the overall approach will be more robust to corrupted data than the Gaussian models where all weights would be deterministic and equal to 0.5.

Unfortunately, the weights $w^{(i)}$ cannot be computed analytically, therefore we need to resort to approximate methods. In this study, we develop a Markov chain Monte Carlo (MCMC) method to approximately compute the weights, where we approximate the intractable expectations with a finite sample average, given as follows: $w_{n,t}^{(i)} \approx (1/J) \sum_{j=1}^{J} 1/\phi_{n,t}^{(i,j)}$, where $\phi_{n,t}^{(i,j)}$ are some samples that are ideally drawn from the posterior distribution $p(\phi|x, z^{(i)}, d^{(i)})$. Unfortunately, directly drawing samples from the posterior distribution of $\phi$ is not tractable either, and therefore, we develop a *Metropolis-Hastings* algorithm [28], that asymptotically generates samples from the *target* distribution $p(\phi|\cdot)$ in two steps. In the $j$-th iteration of this

**Algorithm 1** $\alpha$-stable Convolutional Sparse Coding
___
**Require:** Regularization: $\lambda \in \mathbb{R}_+$, Num. atoms: $K$, Atom length: $L$, Num. iterations: $I$ , $J$, $M$
1: **for** $i = 1$ to $I$ **do**
2:     */* E-step: */*
3:     **for** $j = 1$ to $J$ **do**
4:         Draw $\phi_{n,t}^{(i,j)}$ via MCMC (9)
5:     **end for**
6:     $w_{n,t}^{(i)} \approx (1/J) \sum_{j=1}^{J} 1/\phi_{n,t}^{(i,j)}$
7:     */* M-step: */*
8:     **for** $m = 1$ to $M$ **do**
9:         $z^{(i)}$ = L-BFGS-B on (10)
10:        $d^{(i)}$ = L-BFGS-B on the dual of (11)
11:     **end for**
12: **end for**
13: **return** $w^{(I)}, d^{(I)}, z^{(I)}$
___

algorithm, we first draw a random sample for each $n$ and $t$ from the prior distribution (cf. (5)), *i.e.*, $\phi'_{n,t} \sim p(\phi_{n,t})$. We then compute an acceptance probability for each $\phi'_{n,t}$ that is defined as follows:

$$\text{acc}(\phi_{n,t}^{(i,j)} \to \phi'_{n,t}) \triangleq \min\left\{1, p(x_{n,t}|d^{(i)}, z^{(i)}, \phi'_{n,t})/p(x_{n,t}|d^{(i)}, z^{(i)}, \phi_{n,t}^{(i,j)})\right\} \quad (9)$$

where $j$ denotes the iteration number of the MCMC algorithm. Finally, we draw a uniform random number $u_{n,t} \sim \mathcal{U}([0, 1])$ for each $n$ and $t$. If $u_{n,t} < \text{acc}(\phi_{n,t}^{(i)} \to \phi'_{n,t})$, we accept the sample and set $\phi_{n,t}^{(i+1)} = \phi'_{n,t}$; otherwise we reject the sample and set $\phi_{n,t}^{(i+1)} = \phi_{n,t}^{(i)}$. This procedure forms a Markov chain that leaves the target distribution $p(\phi|\cdot)$ invariant, where under mild ergodicity conditions, it can be shown that the finite-sample averages converge to their true values when $J$ goes to infinity [29]. More detailed explanation of this procedure is given in the supplementary document.

**The M-Step:** Given the weights $w_n$ that are estimated during the E-step, the objective of the M-step (7) is to solve a weighted CSC problem, which is much easier when compared to our original problem. This objective function is not jointly convex in $d$ and $z$, yet it is convex if one fix either $d$ or $z$. Here, similarly to the vanilla CSC approaches [9, 10], we develop a *block coordinate descent* strategy, where we solve the problem in (7) for either $d$ or $z$, by keeping respectively $z$ and $d$ fixed. We first focus on solving the problem for $z$ while keeping $d$ fixed, given as follows:

$$\min_z \sum_{n=1}^{N} \left( \|\sqrt{w_n} \odot (x_n - \sum_{k=1}^{K} D^k \bar{z}_n^k)\|_2^2 + \lambda \sum_k \|z_n^k\|_1 \right) \quad \text{s.t. } z_n^k \geq 0, \forall n, k \quad . \quad (10)$$

Here, we expressed the convolution of $d^k$ and $z_n^k$ as the inner product of the zero-padded activations $\bar{z}_n^k \triangleq [(z_n^k)^\top, 0 \cdots 0]^\top \in \mathbb{R}_+^T$, with a Toeplitz matrix $D^k \in \mathbb{R}^{T \times T}$, that is constructed from $d^k$. The matrices $D^k$ are never constructed in practice, and all operations are carried out using convolutions. This problem can be solved by various constrained optimization algorithms. Here, we choose the quasi-Newton L-BFGS-B algorithm [30] with a box constraint: $0 \leq z_{n,t}^k \leq \infty$. This approach only requires the simple computation of the gradient of the objective function with respect to $z$ (*cf.* supplementary material). Note that, since each trial is independent from each other, we can solve this problem for each $z_n$ in parallel.

We then solve the problem for the atoms $d$ while keeping $z$ fixed. This optimization problem turns out to be a constrained weighted least-squares problem. In the non-weighted case, this problem can be solved either in the time domain or in the Fourier domain [10–12]. The Fourier transform simplifies the convolutions that appear in least-squares problem, but it also induces several difficulties, such as that the atom $d_k$ have to be in a finite support $L$, an important issue ignored in the seminal work of [10] and addressed with an ADMM solver in[11, 12]. In the weighted case, it is not clear how to solve this problem in the Fourier domain. We thus perform all the computations in the time domain.

Following the traditional filter identification approach [31], we need to embed the one-dimensional signals $z_n^k$ into a matrix of delayed signals $Z_n^k \in \mathbb{R}^{T \times L}$, where $(Z_n^k)_{i,j} = z_{n,i+j-L+1}^k$ if $L - 1 \leq$

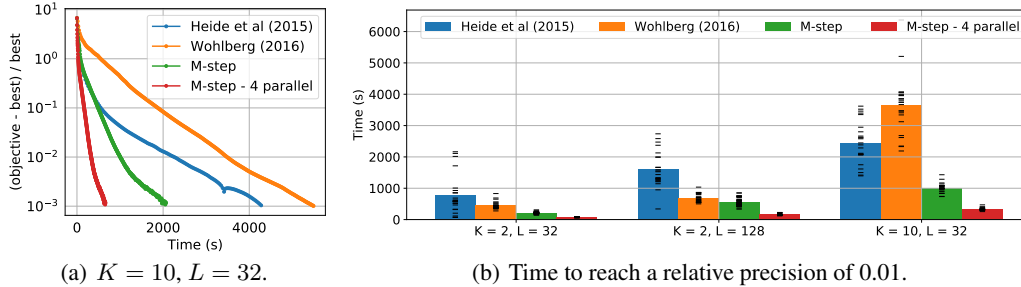

(a) $K = 10, L = 32$.

(b) Time to reach a relative precision of 0.01.

Figure 2: Comparison of state-of-the-art methods with our approach. (a) Convergence plot with the objective function relative to the obtained minimum, as a function of computational time. (b) Time taken to reach a relative precision of $10^{-2}$, for different settings of $K$ and $L$.

$i + j < T$ and 0 elsewhere. Equation (1) then becomes:

$$\min_d \sum_{n=1}^{N} \| \sqrt{w_n} \odot (x_n - \sum_{k=1}^{K} Z_n^k d^k) \|_2^2, \quad \text{s.t.} \ \|d^k\|_2^2 \le 1 \ . \tag{11}$$

Due to the constraint, we must resort to an iterative approach. The options are to use (accelerated) projected gradient methods such as FISTA [32] applied to (11), or to solve a dual problem as done in [10]. The dual is also a smooth constraint problem yet with a simpler positivity box constraint (*cf.* supplementary material). The dual can therefore be optimized with L-BFGS-B. Using such a quasi-Newton solver turned out to be more efficient than any accelerated first order method in either the primal or the dual (*cf.* benchmarks in supplementary material).

Our entire EM approach can be summarized in the Algorithm 1. Note that during the alternating minimization, thanks to convexity we can warm start the $d$ update and the $z$ update using the solution from the previous update. This significantly speeds up the convergence of the L-BFGS-B algorithm, particularly in the later iterations of the overall algorithm.

## 4 Experiments

In order to evaluate our approach, we conduct several experiments on both synthetic and real data. First, we show that our proposed optimization scheme for the M-step provides significant improvements in terms of convergence speed over the state-of-the-art CSC methods. Then, we provide empirical evidence that our algorithm is more robust to artifacts and outliers than three competing CSC methods [6, 7, 12]. Finally, we consider LFP data, where we illustrate that our algorithm can reveal interesting properties in electrophysiological signals without supervision, even in the presence of severe artifacts. The source code is publicly available at `https://alphacsc.github.io/`.

**Synthetic simulation setup:** In our synthetic data experiments, we simulate $N$ trials of length $T$ by first generating $K$ zero mean and unit norm atoms of length $L$. The activation instants are integers drawn from a uniform distribution in $[\![0, T - L]\!]$. The amplitude of the activations are drawn from a uniform distribution in $[0, 1]$. Atoms are activated only once per trial and are allowed to overlap. The activations are then convolved with the generated atoms and summed up as in (1).

**M-step performance:** In our first set of synthetic experiments, we illustrate the benefits of our M-step optimization approach over state-of-the-art CSC solvers. We set $N = 100$, $T = 2000$ and $\lambda = 1$, and use different values for $K$ and $L$. To be comparable, we set $\alpha = 2$ and add Gaussian noise to the synthesized signals, where the standard deviation is set to 0.01. In this setting, we have $w_{n,t} = 1/2$ for all $n, t$, which reduces the problem to a standard CSC setup. We monitor the convergence of ADMM-based methods by Heide et al. [11] and Wohlberg [12] against our M-step algorithm, using both a single-threaded and a parallel version for the $z$-update. As the problem is non-convex, even if two algorithms start from the same point, they are not guaranteed to reach the same local minimum[2]. Hence, for a fair comparison, we use a multiple restart strategy with averaging across 24 random seeds.

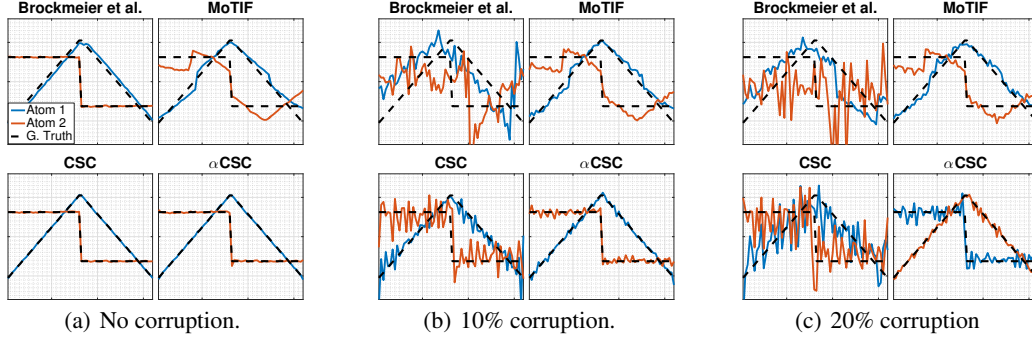

(a) No corruption.  (b) 10% corruption.  (c) 20% corruption

Figure 3: Simulation to compare state-of-the-art methods against $\alpha$CSC.

During our experiments we have observed that the ADMM-based methods do not guarantee the feasibility of the iterates. In other words, the norms of the estimated atoms might be greater than 1 during the iterations. To keep the algorithms comparable, when computing the objective value, we project the atoms to the unit ball and scale the activations accordingly. To be strictly comparable, we also imposed a positivity constraint on these algorithms. This is easily done by modifying the soft-thresholding operator to be a rectified linear function. In the benchmarks, all algorithms use a single thread, except "M-step - 4 parallel" which uses 4 threads during the $z$ update.

In Fig. 2, we illustrate the convergence behaviors of the different methods. Note that the y-axis is the precision relative to the objective value obtained upon convergence. In other words, each curve is relative to its own local minimum (see supplementary document for details). In the right subplot, we show how long it takes for the algorithms to reach a relative precision of $0.01$ for different settings (*cf.* supplementary material for more benchmarks). Our method consistently performs better and the difference is even more striking for more challenging setups. This speed improvement on the M-step is crucial for us as this step will be repeatedly executed.

**Robustness to corrupted data:** In our second synthetic data experiment, we illustrate the robustness of $\alpha$CSC in the presence of corrupted observations. In order to simulate the likely presence of high amplitude artifacts, one way would be to directly simulate the generative model in (3). However, this would give us an unfair advantage, since $\alpha$CSC is specifically designed for such data. Here, we take an alternative approach, where we corrupt a randomly chosen fraction of the trials (10% or 20%) with strong Gaussian noise of standard deviation $0.1$, *i.e.* one order of magnitude higher than in a regular trial. We used a regularization parameter of $\lambda = 0.1$. In these experiments, by CSC we refer to $\alpha$CSC with $\alpha = 2$, that resembles using only the M-step of our algorithm with deterministic weights $w_{n,t} = 1/2$ for all $n$, $t$. We used a simpler setup where we set $N = 100$, $T = 512$, and $L = 64$. We used $K = 2$ atoms, as shown in dashed lines in Fig. 3.

For $\alpha$CSC, we set the number of outer iterations $I = 5$, the number of iterations of the M-step to $M = 50$, and the number of iterations of the MCMC algorithm to $J = 10$. We discard the first 5 samples of the MCMC algorithm as burn-in. To enable a fair comparison, we run the standard CSC algorithm for $I \times M$ iterations, i.e. the *total* number of M-step iterations in $\alpha$CSC. We also compared $\alpha$CSC against competing state-of-art methods previously applied to neural time series: Brockmeier and Príncipe [7] and MoTIF [6]. Starting from multiple random initializations, the estimated atoms with the smallest $\ell_2$ distance with the true atoms are shown in Fig. 3.

In the artifact-free scenario, all algorithms perform equally well, except for MoTIF that suffers from the presence of activations with varying amplitudes. This is because it aligns the data using correlations before performing the eigenvalue decomposition, without taking into account the strength of activations in each trial. The performance of Brockmeier and Príncipe [7] and CSC degrades as

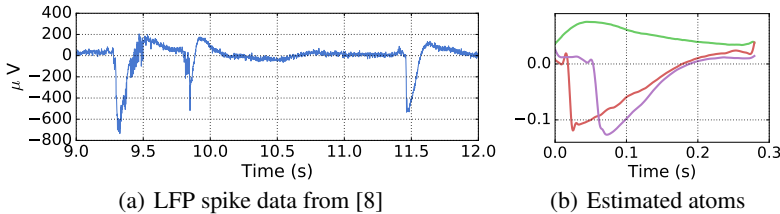

(a) LFP spike data from [8]       (b) Estimated atoms

Figure 4: Atoms learnt by $\alpha$CSC on LFP data containing epileptiform spikes with $\alpha = 2$.

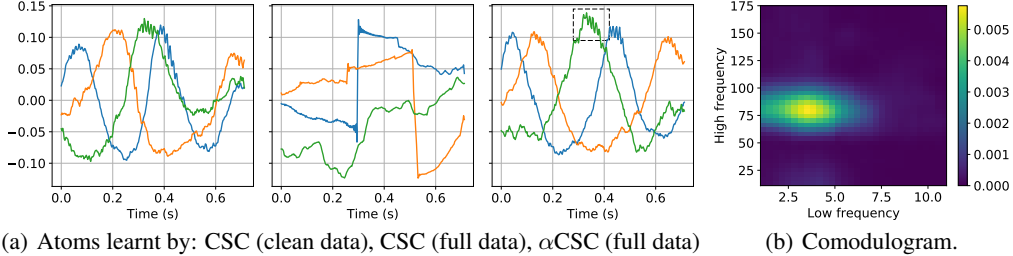

(a) Atoms learnt by: CSC (clean data), CSC (full data), $\alpha$CSC (full data)   (b) Comodulogram.

Figure 5: (a) Three atoms learnt from a rodent striatal LFP channel, using CSC on cleaned data, and both CSC and $\alpha$CSC on the full data. The atoms capture the cross-frequency coupling of the data (dashed rectangle). (b) Comodulogram presents the cross-frequency coupling intensity computed between pairs of frequency bands on the entire cleaned signal, following [37].

the level of corruption increases. On the other hand, $\alpha$CSC is clearly more robust to the increasing level of corruption and recovers reasonable atoms even when 20% of the trials are corrupted.

**Results on LFP data** In our last set of experiments, we consider real neural data from two different datasets. We first applied $\alpha$CSC on an LFP dataset previously used in [8] and containing epileptiform spikes as shown in Fig. 4(a). The data was recorded in the rat cortex, and is free of artifact. Therefore, we used the standard CSC with our optimization scheme, (i.e. $\alpha$CSC with $\alpha = 2$). As a standard preprocessing procedure, we applied a high-pass filter at $1\,\mathrm{Hz}$ in order to remove drifts in the signal, and then applied a tapered cosine window to down-weight the samples near the edges. We set $\lambda = 6$, $N = 300$, $T = 2500$, $L = 350$, and $K = 3$. The recovered atoms by our algorithm are shown in Fig. 4(b). We can observe that the estimated atoms resemble the spikes in Fig. 4(a). These results show that, without using any heuristics, our approach can recover similar atoms to the ones reported in [8], even though it does not make any assumptions on the shapes of the waveforms, or initializes the atoms with template spikes in order to ease the optimization.

The second dataset is an LFP channel in a rodent striatum from [35]. We segmented the data into 70 trials of length 2500 samples, windowed each trial with a tapered cosine function, and detrended the data with a high-pass filter at $1\,\mathrm{Hz}$. We set $\lambda = 10$, initialized the weights $w_n$ to the inverse of the variance of the trial $x_n$. Atoms are in all experiments initialized with Gaussian white noise.

As opposed to the first LFP dataset, this dataset contains strong artifacts, as shown in Fig. 1(b). In order to be able to illustrate the potential of CSC on this data, we first *manually* identified and removed the trials that were corrupted by artifacts. In Fig. 5(a), we illustrate the estimated atoms with CSC on the manually-cleaned data. We observe that the estimated atoms correspond to canonical waveforms found in the signal. In particular, the high frequency oscillations around $80\,\mathrm{Hz}$ are modulated in amplitude by the low-frequency oscillation around $3\,\mathrm{Hz}$, a phenomenon known as cross-frequency coupling (CFC) [36]. We can observe this by computing a comodulogram [37] on the entire signal (Fig. 5(b)). This measures the correlation between the amplitude of the high frequency band and the phase of the low frequency band.

Even though CSC is able to provide these excellent results on the cleaned data set, its performance heavily relies on the manual removal of the artifacts. Finally, we repeated the previous experiment on the full data, without removing the artifacts and compared CSC with $\alpha$CSC, where we set $\alpha = 1.2$. The results are shown in the middle and the right sub-figures of Fig. 5(a). It can be observed that in the presence of strong artifacts, CSC is not able to recover the atoms anymore. On the contrary, we observe that $\alpha$CSC can still recover atoms as observed in the artifact-free regime. In particular, the cross-frequency coupling phenomenon is still visible.

## 5   Conclusion

We address the present need in the neuroscience community to better capture the complex morphology of brain waves. Our approach is based on a probabilistic formulation of a CSC model. We propose an inference strategy based on MCEM to deal efficiently with heavy tailed noise and take into account the polarity of neural activations with a positivity constraint. Our problem formulation allows the use of fast quasi-Newton methods that outperform previously proposed state-of-the-art ADMM-based algorithms, even when not making use of our parallel implementation. Results on LFP data demonstrate that such algorithms can be robust to the presence of transient artifacts in data and reveal insights on neural time-series without supervision.

# 6 Acknowledgement

The work was supported by the French National Research Agency grants ANR-14-NEUC-0002-01, ANR-13-CORD-0008-02, and ANR-16-CE23-0014 (FBIMATRIX), as well as the ERC Starting Grant SLAB ERC-YStG-676943.

## Footnotes

[1]Note that the positivity constraint on the activations is equivalent to an exponential prior for the regularization term rather than the more common Laplacian prior.

[2]Note that the M-step can be viewed as a biconvex problem, for which global convergence guarantees can be shown under certain assumptions [33, 34]. However, we have observed that it is required to use multiple restarts even for vanilla CSC, implying that these assumptions are not satisfied in this particular problem.

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
