[Supplementary Material · nips_2017_supp.pdf]

# Learning the Morphology of Brain Signals Using Alpha-Stable Convolutional Sparse Coding
### SUPPLEMENTARY DOCUMENT

**Mainak Jas**
LTCI, Telecom ParisTech,
Université Paris-Saclay,
Paris, France
mainak.jas@
telecom-paristech.fr

**Tom Dupré La Tour**
LTCI, Telecom ParisTech,
Université Paris-Saclay,
Paris, France
tom.duprelatour@
telecom-paristech.fr

**Umut Şimşekli**
LTCI, Telecom ParisTech,
Université Paris-Saclay,
Paris, France
umut.simsekli@
telecom-paristech.fr

**Alexandre Gramfort**
INRIA,
Université Paris Saclay,
Saclay, France
alexandre.gramfort@inria.fr

## 1   Details of the E-Step

Computing the weights that are required in the M-step requires us to compute the expectation of $\frac{1}{\phi_{n,t}}$ under the posterior distribution $p(\phi_{n,t}|x,d,z)$, which is not analytically available.

Monte Carlo methods are numerical techniques that can be used to approximately compute the expectations of the form:

$$\mathbb{E}[f(\phi_{n,t})] = \int f(\phi_{n,t})\pi(\phi_{n,t})d\phi_{n,t} \approx \frac{1}{J}\sum_{j=1}^{J} f(\phi_{n,t}^{(j)}) \tag{S1}$$

where $\phi_{n,t}^{(j)}$ are some samples drawn from $\pi(\phi_{n,t}) \triangleq p(\phi_{n,t}|x,d,z)$ and $f(\phi) = 1/\phi$ in our case. However, in our case, sampling directly from $\pi(\phi_{n,t})$ is also unfortunately intractable.

MCMC methods generate samples from the target distribution $\pi(\phi_{n,t})$ by forming a Markov chain, whose stationary distribution is $\pi(\phi_{n,t})$, so that $\pi(\phi_{n,t}) = \int \mathcal{T}(\phi_{n,t}|\phi'_{n,t})p(\phi'_{n,t})d\phi'_{n,t}$, where $\mathcal{T}$ denotes the transition kernel of the Markov chain.

In this study, we develop a Metropolis-Hastings (MH) algorithm, that implicitly forms a transition kernel. The MH algorithm generates samples from a target distribution $\pi(\phi_{n,t})$ in two steps. First, it generates a random sample $\phi'_{n,t}$ from a *proposal* distribution $\phi'_{n,t} \sim q(\phi'_{n,t}|\phi_{n,t}^{(j)})$, then computes an acceptance probability $\mathrm{acc}(\phi_{n,t}^{(j)} \rightarrow \phi'_{n,t})$ and draws a uniform random number $u \sim \mathcal{U}([0,1])$. If $u < \mathrm{acc}(\phi_{n,t}^{(j)} \rightarrow \phi'_{n,t})$, it accepts the sample and sets $\phi_{n,t}^{(j+1)} = \phi'_{n,t}$; otherwise it rejects the sample and sets $\phi_{n,t}^{(j+1)} = \phi_{n,t}^{(j)}$. The acceptance probability is given as follows

$$\mathrm{acc}(\phi_{n,t} \rightarrow \phi'_{n,t}) = \min\Big\{1, \frac{q(\phi_{n,t}|\phi'_{n,t})\pi(\phi'_{n,t})}{q(\phi'_{n,t}|\phi_{n,t})\pi(\phi_{n,t})}\Big\} = \min\Big\{1, \frac{q(\phi_{n,t}|\phi'_{n,t})p(x_{n,t}|\phi'_{n,t},d,z)p(\phi'_{n,t})}{q(\phi'_{n,t}|\phi_{n,t})p(x_{n,t}|\phi_{n,t},d,z)p(\phi_{n,t})}\Big\}$$
$$\tag{S2}$$

where the last equality is obtained by applying the Bayes rule on $\pi$.

The acceptance probability requires the prior distribution of $\phi$ to be evaluated. Unfortunately, this is intractable in our case since this prior distribution is chosen to be a positive $\alpha$-stable distribution whose PDF does not have an analytical form. As a remedy, we choose the prior distribution of $\phi_{n,t}$ as the proposal distribution, such $q(\phi_{n,t}|\phi'_{n,t}) = p(\phi_{n,t})$. This enables us to simplify the acceptance probability. Accordingly, for each $\phi_{n,t}$, we have the following acceptance probability:

$$\text{acc}(\phi_{n,t}^{(i,j)} \to \phi'_{n,t}) \triangleq \min\left\{1, \exp(\log \phi_{n,t}^{(i,j)} - \log \phi'_{n,t})/2 + (x_{n,t} - \hat{x}_{n,t}^{(i)})^2(1/\phi_{n,t}^{(i,j)} - 1/\phi'_{n,t})\right\}. \tag{S3}$$

Thanks to the simplification, this probability is tractable and can be easily computed.

## 2 Details of the M-Step

### 2.1 Solving for the activations

In the M-step, we optimize (10) to find the activations $z_n^{(i)}$ of each trial $n$ independently. To keep the notation simple, we will drop the index for the iteration number $i$ of the EM algorithm.

First, this equation can be rewritten by concatenating the Toeplitz matrices for the $K$ atoms into a big matrix $D = [D^1, D^2, ..., D^K] \in \mathbb{R}^{T \times KT}$ and the activations for different atoms into a single vector $\bar{z}_n = [(\bar{z}_n^1)^\top, (\bar{z}_n^2)^\top, ..., (\bar{z}_n^K)^\top]^\top \in \mathbb{R}_+^{KT}$ where $(\cdot)^\top$ denotes the transposition operation. Recall that $\bar{z}_n^k$ is a zero-padded version of $z_n^k$. This leads to a simpler formulation and the objective function $\mathcal{L}(d, z)$:

$$\mathcal{L}(d, z) = \sum_{n=1}^{N} \frac{1}{2}\|\sqrt{w_n} \odot (x_n - D\bar{z}_n)\|_2^2 + \lambda \mathbb{1}^\top \bar{z}_n \ , \tag{S4}$$

where $\mathbb{1} \in \mathbb{R}^{KT}$ is a vector of ones.

The derivative w.r.t. $z_n$ now reads:

$$\frac{\partial \mathcal{L}(d, z)}{\partial \bar{z}_n} = D^\top(w_n \odot (x_n - D\bar{z}_n)) + \lambda \mathbb{1}^\top \ . \tag{S5}$$

In practice, this big matrix $D$ is never assembled and all operations are carried out using convolutions. Note also that we do not update the zeros from the padding in $\bar{z}_n^k$. Now that we have the gradient, the activations can be estimated using a efficient quasi-Newton solver such as L-BFGS-B, taking into account the box posititivy constraint $0 \le z_n \le \infty$.

For each trial, one iteration costs $\mathcal{O}(LKT)$.

### 2.2 Solving for the atoms

In the M-step, we optimize (11) to find the atoms $d^k$. As when solving for the activations $z_n$, we can remove the summation over the atoms by concatenating the delayed matrices into $Z_n = [Z_n^1, Z_n^2, ..., Z_n^K] \in \mathbb{R}^{T \times KL}$ and $d = [(d^1)^\top, (d^2)^\top, ..., (d^K)^\top]^\top \in \mathbb{R}^{KL}$. This leads to the simpler formulation:

$$\min_d \sum_{n=1}^{N} \frac{1}{2}\|\sqrt{w_n} \odot (x_n - Z_n d)\|_2^2, \quad \text{s.t. } \|d^k\|_2^2 \le 1 \ . \tag{S6}$$

The Lagrangian of this problem is given by:

$$g(d, \beta) = \sum_{n=1}^{N} \frac{1}{2}\|\sqrt{w_n} \odot (x_n - \sum_{k=1}^{K} Z_n^k d^k)\|_2^2 + \sum_k \beta^k(\|d^k\|_2^2 - 1) \quad \text{s.t. } \beta^k \ge 0 \ , \tag{S7}$$

where $\beta = (\beta^1, \beta^2, ..., \beta^K)$ are the dual variables. Therefore, the dual problem is:

$$\min_d g(d, \beta) = g(d^*, \beta) \tag{S8}$$

where $d^*$, the primal optimal, is given by:

$$d^* = (\sum_{n=1}^{N} Z_n^\top(w_n \odot Z_n) + \bar{\beta})^{-1} \sum_{n=1}^{N} (w_n \odot Z_n)^\top x_n \tag{S9}$$

with $\bar{\beta} = \mathrm{diag}([\mathbb{1}\beta^1, \mathbb{1}\beta^2, ..., \mathbb{1}\beta^K]) \in \mathbb{R}^{KL}$ with $\mathbb{1} \in \mathbb{R}^L$. The gradient for the dual variable $\beta^k$ is given by:

$$\frac{\partial g(d^*, \beta)}{\partial \beta^k} = \|d^{*k}\|_2^2 - 1, \tag{S10}$$

with $d^{*k}$ computed from (S9). We can solve this iteratively using again L-BFGS-B taking into account the positivity constraint $\beta^k \geq 0$ for all $k$. What we have described so far solves for all the atoms simultaneously. However, it is also possible to estimate the atoms sequentially one at a time using a block coordinate descent (BCD) approach, as in the work of [1]. In each iteration of the BCD algorithm, a residual $r_n^k$ is computed as given by:

$$r_n^k = x_n - \sum_{k' \neq k} Z_n^{k'} d^{k'} \tag{S11}$$

and correspondingly subproblem S6 becomes:

$$\min_{d^k} \sum_{n=1}^{N} \frac{1}{2} \|\sqrt{w_n} \odot (r_n^k - Z_n^k d^k)\|_2^2, \quad \text{s.t. } \|d^k\|_2^2 \leq 1, \ . \tag{S12}$$

which is solved in the same way as subproblem S6. Now, in the simultaneous case, we construct one linear problem in $\mathcal{O}(L^2 K^2 TN)$ and one iteration costs $\mathcal{O}(L^3 K^3)$. However, in the BCD strategy, we construct $K$ linear problems in $\mathcal{O}(L^2 TN)$ and one iteration costs only $\mathcal{O}(L^3)$. Interestingly, when the weights $w_n$ are all identical, we can use the fact that for one atom $k$, the matrix $\sum_{i=1}^{N} (Z_i^k)^T Z_i^k$ is Toeplitz. In this case, we can construct $K$ linear problems in only $\mathcal{O}(LTN)$ and one iteration costs only $\mathcal{O}(L^2)$.

For the benefit of the reader, we summarize the complexity of the M-step in Table 1. We note $p$ and $q$ the number of iterations in the L-BFGS-B methods for the activations update and atoms update.

| Method | Complexity |
|---|---|
| Solving activations $z$ | $p \min(L, \log(T)) KTN$ |
| Solving atoms $d$ | $L^2 K^2 TN + qL^3 K^3$ |
| Solving atoms $d$ (sequential) | $LKTN + qL^2 K$ |

Table 1: Complexity analysis of the M-step, where $p$ and $q$ are the number of iterations in the L-BFGS-B solvers for the activations and atoms updates.

# 3 Additional Experiments: M-step speed benchmark

## 3.1 Comparison with state of the art

(a) $K = 2$, $L = 32$.  (b) $K = 2$, $L = 128$.  (c) $K = 10$, $L = 32$.

Figure S1: Convergence speed of the relative objective function. The y-axis shows the objective function relative to the obtained minimum for each run: $(f(x) - f(x^*))/f(x^*)$. Each curve is the geometrical mean over 24 different random initializations.

Here, we compare convergence plots of our algorithm against a number of state-of-art methods. The details of the experimental setup are described in Section 4. Fig. S1 demonstrates on a variety of

setups the computational advantage of our quasi-Newton approach to solve the M-step. Note that Fig. 2b is in fact a summary of Fig. S1. Indeed, we can verify that ADMM methods converge quickly to a modest accuracy, but take much longer to converge to a high accuracy.[2]

Next, in Fig. S2, we show more traditional convergence plots. In contrast to Fig. 2 or S1 where the relative objective function is shown, here we plot the absolute value of the objective function. We can now verify that each of the methods have indeed converged to their respective local minimum. Of course, owing to the non-convex nature of the problem, they do not necessarily converge to the same local minimum.

(a) $K = 2, L = 32$.  (b) $K = 2, L = 128$.  (c) $K = 10, L = 32$.

Figure S2: Convergence of the objective function as a function of time. The y-axis shows the absolute objective function $f(x)$. Each curve is the mean over 24 different random initializations.

## 3.2    Comparison of solver for the activations subproblem

Finally, we compare convergence plots of our algorithm using different solvers for the $z$-update: ISTA, FISTA, and L-BFGS-B. The rationale for choosing a quasi-Newton solver for the $z$-update becomes clear in Fig. S3 as the L-BFGS-B solver turns out to be computationally advantageous on a variety of setups.

(a) $K = 2, L = 32$.  (b) $K = 2, L = 128$.  (c) $K = 10, L = 32$.

Figure S3: Convergence speed of the relative objective function. The y-axis shows the objective function relative to the obtained minimum for each run: $(f(x) - f(x^*))/f(x^*)$. Each curve is the geometrical mean over 24 different random initializations.