[Reviews · NeurIPS 2017]

Reviewer 1



In this paper, the authors propose a novel probabilistic convolutional sparse coding model (alpha-stable CSC) for learning shift-invariant atoms from raw neural signals. They propose an inference strategy based on Monte Carlo EM to deal efficiently with heavy tailed noise and take into account the polarity of neural activations with a positivity constraint. The formulation allows the use of fast quasi-Newton methods for the M-step which outperform previously proposed state-of-the-art ADMM based algorithms. The experiment results on LFP data demonstrate that such algorithms can be robust to the presence of transient artifacts in data and reveal insights on neural time-series without supervision. The main contribution of this study is extending a general (alpha-stable distribution) framework, comparing to the Gaussian distribution. From experiment point of view, the proposed approach is very useful. However, there are a few comments and suggestions for the authors to consider: (1) Figure 1(a) only demonstrates the PDFs of alpha-stable distributions characterized by alpha and beta. Please try to describe the characters of the scale parameter and location parameter. (2) As written in the paper, “As an important special case of the alpha-stable distributions, we obtain the Gaussian distribution when alpha= 2 and beta = 0”, the authors emphasize that the current CSC models might be limited as they consider an L2 reconstruction error. While the extending model is more robust to corrupted data than the Gaussian models. However, I wonder if the experiments should include Gaussian Mixture Model as a fair comparison. (3) How to determine the parameters (i.e., lambda, N, T, L, K), respectively, in synthetic and LEP data experiments?

Reviewer 2



The paper proposes a probabilistic variant of Convolutional Sparse Coding using alpha Stable distributions. Overall the paper is well written, and the method appears to be novel. The authors develop an EM approach to solving the problem, where the E step corresponds to finding the distributional parameters and an M step that corresponds to standard (non-probabilistic) CSC. This appears to improve robustness. Another plus for this paper is that source code has been provided. - L62: "l2 reconstruction error, which corresponds to assuming an additive Gaussian noise distribution". I don't think this is true: l2 reconstruction error corresponds to a Gaussian likelihood. A Gaussian noise distribution corresponds to a l2 regulariser - L115-7: "It is easy to verify that the MAP estimate for this probabilistic model, i.e. maxd,z log p(d, z|x), is identical to the original optimization problem defined in (1)." I've Not come across the 1 norm being the MAP estimate of the exponential distribution before (normally Laplace (double exponential)). Could you explain how this is the case? - eq 3: Should mention that only symmetric stable distributions are being considered - L139-140: "It can be easily shown that both formulations of the CSC model are identical by marginalizing the joint distribution". Not immediately obvious - can this be included in the supplemental? - L157 not obvious where the weights come from

Reviewer 3



The paper presents a new sparse coding method by using alpha-stable sampling distributions (instead of Gaussians) when sampling convolutional sparse codes. This proposed model is shown to approximate the heavy-tailed neural data distributions. A sampling based inference algorithm is presented and evaluated on neural time-series data. All told this is a good contribution. The necessity of modeling heavy-tailed data and noise is critical in neuroscience applications where, in certain cases, these tails contain the discriminatory signals among different groups/subsets/classes of the data. Below are some pointers to improve the paper. 1. The authors propose a EM and MH based sampling learning/inference for the proposed model. However, CSC can be posed as efficient bi-convex problem which gives guarantees (unlike the EM-style local minima). Does the proposed model entail a more biconvex type learning? If not, this needs to explained clearly (and related work needs to be improved). If so, then why is that modeling not considered? Does taking the sampling route tie-up to some properties of alpha-stable distributions? These modeling aspects needs to be explained. 2. It seems that the influence/choice of alpha is not explicit in the sampling scheme. Since the basis of the paper is that alpha is not going to be equal to 2 (i.e., gaussian is not the correct choice), some evaluations justifying this in practice needs to be reported (e.g., estimating the alpha that gave rise to the correct bases from Fig 5; beyond simply setting it to some fixed value like 1.2). More generally influence of all hyper-parameters needs to be addressed.